# A Stitch in Time Saves Nine: Proactive Self-Refinement for Language Models

**Jinyi Han**[♡]**, Xinyi Wang** [♢]**, Haiquan Zhao**[♠]**, Tingyun li**[♢]**, Zishang Jiang** [♢]**, Sihang Jiang** [♠]**,
Jiaqing Liang** [♢]**, Xin Lin** [♡]**, Weikang Zhou** [♣]**, Zeye Sun** [♣]**, Fei Yu** [♣]**, Yanghua Xiao**[♡, ♠] [*]

[♡]Shanghai Institute of Artificial Intelligence for Education, East China Normal University
[♢]School of Data Science, Fudan University
[♠]College of Computer Science and Artificial Intelligence, Fudan University [♣] Ant Group

## Abstract

Recent advances in self-refinement have demonstrated significant potential for improving the outputs of large language models (LLMs) through iterative refinement. However, most existing self-refinement methods rely on a reactive process with a fixed number of iterations, making it difficult to determine the optimal timing and content of refinement based on the evolving generation context. Inspired by the way humans dynamically refine their thoughts during execution, we propose ProActive Self-Refinement (PASR), a novel method that enables LLMs to refine their outputs during the generation process. Unlike methods that regenerate entire responses, PASR proactively decides whether, when, and how to refine based on the model's internal state and evolving context. We conduct extensive experiments on a diverse set of 10 tasks to evaluate the effectiveness of PASR. Experimental results show that PASR significantly enhances problem-solving performance. In particular, on Qwen3-8B, PASR reduces average token consumption by 41.6% compared to standard generation, while also achieving an 8.2% improvement in accuracy. Our code and baselines used in the paper are available in the GitHub [1].

## 1 Introduction

Self-refinement, as a fundamental cognitive capacity, is essential for effective problem-solving in humans. It involves actively monitoring one's thought processes, identifying and correcting errors, and iteratively adjusting responses and behaviors (Dewey, 1986; Kuhl & Beckmann, 2012). Its significance in human intelligence highlights a promising direction for developing more autonomous and robust AI agents. Inspired by this powerful cognitive process, recent work has applied the self-refinement to Large Language Models (LLMs).

Existing self-refinement methods for LLMs typically follow **patch-after-failure (post-hoc)** paradigm, where an initial response is generated and then iteratively improved based on feedback through multiple rounds of refinement iterations(Madaan et al., 2023; Welleck et al., 2023; Huang et al., 2024; Ganguli et al., 2023b). Broadly, these methods fall into two categories. The first employs carefully crafted prompts to elicit self-refinement behaviors, often by explicitly instructing it to correct or refine its previous outputs (Ganguli et al., 2023a; Olausson et al., 2024; 2023a). The second leverages Supervised Fine-Tuning (SFT) on synthetic datasets that pair suboptimal responses with improved versions, training the model to refine its outputs automatically (Havrilla et al., 2024; Du et al., 2025). (Tong et al., 2024; Xie et al., 2025; An et al., 2024).

While these post-hoc self-refinement methods have improved performance on various tasks, they remain fundamentally reactive and lack the ability to proactively determine **whether, when and how** to perform refinement. *(Whether:)* these methods are often applied blindly after initial generation, requiring multiple iterations whose optimal number is unclear and usually demands extensive tuning (Du et al., 2025; Madaan et al., 2023). *(When:)* errors arising during initial generation can propagate through subsequent steps (Gan et al., 2025; Bachmann & Nagarajan, 2024), making later correction

---

[*] Corresponding author
[1]https://github.com/JinyiHan99/Proactive-Self-Refine-in-LLMs/

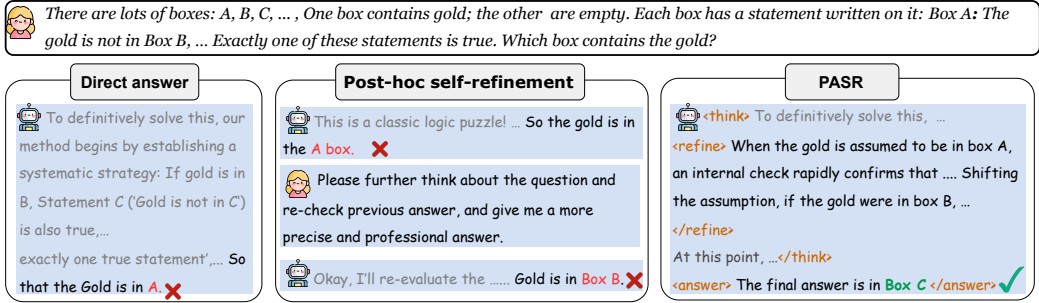

Figure 1: Comparison between the post-hoc refinement method (middle) and our proposed PASR (right). The post-hoc refinement method iteratively refines its initial answer. In contrast, PASR proactively refines its reasoning process during the generation.

more difficult. *(How:)* these methods rely heavily on external feedback mechanisms, such as tool-assisted evaluations and auxiliary models (Gou et al., 2024; Xie et al., 2025; Chen et al., 2024), and inappropriate feedback even degrade the performance (Huang et al., 2024).

It is crucial to equip LLMs with **proactive self-refinement** capabilities during generation, allowing models to autonomously determine the appropriate timing and content for refinement based on the evolving context. While advanced reasoning models like DeepSeek-R1 (Guo et al., 2025) and OpenAI-o1 (Jaech et al., 2024) demonstrate some in-process refinement behaviors, these mechanisms are neither explicitly designed for proactive self-refinement nor systematically evaluated for their impact on output quality. Furthermore, the underlying mechanisms driving these refinements remain unclear, limiting our understanding of how to develop more effective self-refinement capabilities in LLMs.

A straightforward approach for equipping LLMs with proactive self-refinement is training them on demonstrations of adaptive refinement behavior. However, this method faces two significant challenges. First, constructing such demonstration data is non-trivial, as defining the optimal timing for refinement during generation is impractical and distilling it from advanced LLMs is not feasible. Second, merely imitating these demonstrations is insufficient for the model to truly acquire the capability (Kumar et al., 2025; Wang et al., 2025). Models struggles to generalize adaptive self-refinement behavior to unseen tasks, and in some cases, their performance even deteriorates.

Therefore, we propose **P**ro**A**ctive **S**elf-**R**efinement (PASR), a Reinforcement Learning (RL) method that trains LLMs to adaptively refine their outputs during generation. Unlike post-hoc refinement, which is applied after generation based on predefined rules, PASR leverages on-policy rollouts to explore whether, when, and how to refine, conditioned on the task and generation state (Figure 1). In contrast to SFT, RL shapes the model's behavior through reward signals (Lee et al., 2024; Yuan et al., 2024). A key challenge is defining what counts as an effective refinement. If the rewards are misaligned, the model may either miss important refinement opportunities or make unnecessary modifications to already correct outputs. To address this, we introduce a proxy evaluation strategy that compares refinements against standard outputs, encouraging timely, necessary, and contextually appropriate refinement.

In summary, our main contributions are summarized as follows:

• We formally define proactive self-refinement as a task, allowing models to iteratively decide whether, when, and how to refine.
• We introduce ProActive Self-Refinement (PASR), a reinforcement learning framework that enables LLMs to autonomously refine their outputs during generation.
• We design a comparison-based reward that encourages timely, necessary, and contextually appropriate refinements.
• Extensive experiments show that PASR improves both efficiency and accuracy. Notably, on Qwen3-8B, it reduces token consumption by 41.6% while increasing accuracy by 8.2%, demonstrating the practical effectiveness of proactive self-refinement.

## 2 METHOD

### 2.1 TASK FORMULATION

Unlike existing post-hoc refinement methods, our task is that empowers the model to proactive self-refine its generated content during the generation process. We formalize this in-process refinement behavior as follows:

*Error Correction.* Fixing factual inaccuracies, logical fallacies, or computational mistakes introduced in earlier outputs.

*Information Complement.* Filling in missing yet critical details to ensure completeness and correctness.

*Solution Improvement.* Improving the effectiveness and efficiency of the proposed solution by introducing more advanced strategies or refined representations.

*Task Alignment.* Re-aligning content with the task goal or user intent when divergence is detected.

The model proactively decides whether, when and how to refine previously generated parts of its internal reasoning trace, integrating these updates into its ongoing generation process. This sequential decision-making problem is naturally formulated as a Markov Decision Process (MDP) (Bellman, 1957).

Formally, given an input query $x$, the goal is to generate a final response $y^{'}$. This is achieved through an iterative refinement process that constructs an intermediate generation trace $z = (z_1, z_2, \ldots, z_T)$, where $T$ is the total number of generation tokens. At each timestep $i$ (from 1 to $T$), the model is in the **state** $s_i$, which is determined by the input $x$ and the trace generated $z_{\{1:i-1\}}$ so far. It then takes an **action** $a_i$ chosen from an action space $\mathcal{A}$, which consists of two main types of actions: *Content Generation* $a_{\text{gen}}$ and *Trace Refinement* $a_{\text{refine}}$. The *Content Generation* extends the current line of reasoning. The model produces the next reasoning step and appends it directly to the end of the existing trace, thereby moving the reasoning process forward. The *Trace Refinement* focuses on improving the quality of the already generated trace. Instead of advancing the reasoning, the model inspects previously produced content, identifies potential weaknesses, and generates corrective or explanatory additions to enhance clarity, consistency, or correctness. The sequence of states, actions, and resulting trace segments $((s_1, a_1, z_1), \ldots, (s_T, a_T, z_T))$ constitutes an **observation**. The final response $y^{'}$ is derived from the complete trace $z$. The training objective is to learn the optimal **policy** $\pi$ that maximizes the expected reward of proactive refinement responses. The reward, denoted as $R_{y'}$, reflects the quality of the response resulting from proactive trace refinement. The objective is formalized as:

$$\max_{\pi} \sum_{x} \mathbb{E}_{y' \sim \pi(\cdot|x)} \left[ R_{y'} \right] \tag{1}$$

### 2.2 PASR: PROACTIVE SELF-REFINEMENT VIA RL

In this work, we employ Group Relative Policy Optimization (GRPO) algorithm, a variant of Proximal Policy Optimization (PPO), specifically designed to stabilize training through group-wise advantage normalization. For each query $x$, the policy $\pi_\theta$ samples a group of candidate responses $G_x = \{(y_1^{'}, R_{y_1^{'}}), \cdots, (y_n^{'}, R_{y_n^{'}})\}$. where each pair contains a response and its reward.

We normalize the advantage of each response $y_i^{'}$ in group $G_x$ as:

$$A_i(y_i^{'}|x) = \frac{R_{y_i^{'}} - \mu_x}{\sigma_x + \xi}, \tag{2}$$

where $\mu_x$ and $\sigma_x$ are the mean and standard deviation of rewards in $G_x$, and $\xi$ is a small constant added for numerical stability to avoid division by zero. The GRPO objective function $J_{GRPO}(\theta)$ is formulated to balance reward maximization and policy stability, which is defined as:

$$J_{\text{GRPO}}(\theta) = \mathbb{E}_G \left[ \frac{1}{G} \sum_{i=1}^{G} A_i(y_i'|x) \cdot \min \left( r_i, \text{clip}(r_i, 1 - \epsilon, 1 + \epsilon) \right) - \beta D_{\text{KL}}(\pi_\theta(\cdot|x) \| \pi_{\text{ref}}(\cdot|x)) \right] \tag{3}$$

where $r_i = \frac{\pi_\theta(y_i^{'}|x)}{\pi_{\text{old}}(y_i^{'}|x)}$, $\pi_{\text{old}}$ is the policy before the update. $\epsilon$ is a hyperparameter controlling the clipping range, and $\beta$ weights the KL-divergence penalty. The KL divergence term, $D_{\text{KL}}(\pi_\theta \| \pi_{\text{ref}}) =$

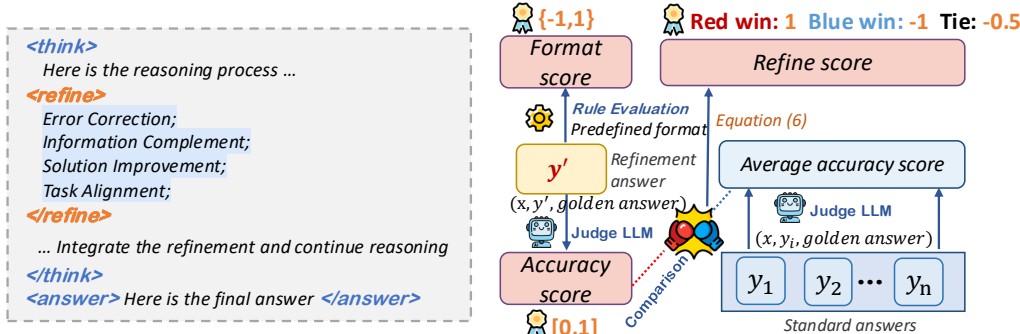

Figure 2: **Left:** Answer format used in PASR. **Right:** Reward design for a generated answer $y'$ during training. The total reward is computed as the sum of the format score, accuracy score, and refinement score, as defined in Equation 7.

$\frac{\pi_{\text{ref}}(y'_i|x)}{\pi_\theta(y'_i|x)} - \log\left(\frac{\pi_{\text{ref}}(y'_i|x)}{\pi_\theta(y'_i|x)}\right) - 1$, enforces proximity to a reference policy $\pi_{\text{ref}}$, thus preventing excessive policy shifts and mitigating the risk of over-optimization.

**PASR Rollout.** To enable the model to autonomously determine both whether, when and how to perform refinement during the generation process, we first design a structured output format guided by a system prompt. The prompt is shown in Table 6.

The system prompt explicitly instructs the model to structure its output using three specialized tags: *<think>*, *<refine>* and *<answer>*, which denote the reasoning trajectory, refinement segments, and final response, respectively. The *<think>* tag encapsulates the model's entire reasoning trajectory. Within this reasoning scope, the *<refine>* tag identifies specific segments where the model revises or improves previously generated content. The *<refine>* tag required to be nested within the *<think>* tag, indicating that refinement is an integral part of the model's reasoning process. After each *<refine>* segment, the model continues reasoning based on the updated content, allowing refinements to directly influence subsequent steps. The model is encouraged to perform recursive refinement, allowing it to invoke the *<refine>* action multiple times during a single generation whenever it deems further improvements beneficial.

The use of these specialized tags enforces a **semantically structured generation process**, guiding the model to clearly distinguish and focus on each phase, including reasoning, refinement, and final response, with each phase serving an explicit functional role. The refinement output format of PASR is illustrated in Figure 2.

## 2.3 Reward Design

Rule-based reward mechanisms have demonstrated strong empirical performance and are widely adopted in RL settings (Dao & Vu, 2025; Shao et al., 2024). In our training framework, we employ a hybrid reward scheme that combines rule-based and model-based evaluation to guide both generation and refinement behavior. Specifically, we define three types of rewards: the format reward $r_{\text{format}}$, the accuracy reward $r_{\text{acc}}$ and the refinement reward $r_{\text{refine}}$.

**Format Reward.** This reward evaluates whether the generated output conforms to predefined structural constraints, defined as follows:

***Constraint 1 (C1):*** the output must include both *<think>* and *<answer>* tag pairs; the *<refine>* tag is optional.

***Constraint 2 (C2):*** if the *<refine>* tag appears, it must be properly nested within the *<think>* tag.

***Constraint 3 (C3):*** the relative order of the three tags must be preserved and cannot be rearranged.

Let $C_i(y') \in {0, 1}$ indicates whether condition $C_i$ is satisfied for a given output $y'$. The format reward $r_{\text{format}}(y')$ is then defined as:

$$r_{format}(y') = 2(C_1(y')\, C_2(y')\, C_3(y')) - 1 \tag{4}$$

This formulation assigns a reward of 1 if and only if all constraints are satisfied; otherwise, a penalty of -1 is applied. The strict binary scheme ensures that only fully well-formed outputs are positively reinforced.

**Accuracy Reward.** It is designed to evaluate the quality and correctness of PASR's generated answers. As our training tasks are drawn from open-domain question, many are inherently ambiguous or under-specified. Consequently, outputs are diverse and expressed in free-form language, making rule-based checks or exact string matching ineffective.

To address this issue, following prior work (Zheng et al., 2023), we employ an advanced LLM as a judge model. The evaluation model is prompted with three components: the original question $x$, the generated answer $y^{'}$ and the reference answer $\hat{y}$. It then outputs a continuous score in the range $[0, 1]$, reflecting the semantic quality and task relevance of the generated response relative to the reference. Let $\mathcal{J}$ denote the judgment function, the accuracy reward $r_{acc}(y^{'})$ is defined as:

$$r_{\text{acc}}(y^{'}) = \mathcal{J}(x, \hat{y}, y^{'}) \tag{5}$$

**Refinement Reward.** It is used to assess whether refinement actions of $y^{'}$ are beneficial and timely. Directly measuring the effectiveness of adaptive self-refinement is challenging, we instead employ a proxy evaluation strategy that assesses refinement quality by **comparing** the refined response $y'$ with a set of standard responses $y$ without refinement. Given the stochastic nature of the model's generation, we sample multiple standard responses to estimate the expected accuracy of the model, denoted as $\bar{r}_{acc}(y)$. The refinement reward is designed according to the follows principles:

***Reward effective refinements.*** A positive reward is assigned if the refined response achieves significantly higher accuracy than the baseline average.

***Penalize harmful refinements.*** A negative reward is given if refinement decreases accuracy relative to the baseline average.

***Discourage unnecessary refinements.*** If the refined response yields comparable accuracy to the baseline average, a small penalty is applied to discourage redundant changes.

Formally, the refinement reward is defined as:

$$r_{refine}(y^{'}) = \begin{cases} 1, r_{acc}(y^{'}) > \bar{r}_{acc}(y) + \zeta \\ -1, r_{acc}(y^{'}) < \bar{r}_{acc}(y) - \zeta \\ -0.5, |r_{acc}(y^{'}) - \bar{r}_{acc}(y)| \leq \zeta \end{cases} \tag{6}$$

Here, $\zeta$ is a tolerance parameter that provides robustness against noise and minor fluctuations. This formulation encourages the model to refine its output only when the refinement yields a measurable gain, while penalizing ineffective or unnecessary modifications.

**Overall Reward.** The final reward for each response generated by $\pi_\theta$ is computed as the sum of the three components.

$$R_{y'} = r_{format}(y^{'}) + r_{acc}(y^{'}) + r_{refine}(y^{'}) \tag{7}$$

Unlike prior approaches that rely solely on binary reward signals, our **fine-grained** reward is designed to encourage meaningful and constructive refinement while explicitly discouraging both excessive and insufficient refinement.

## 3 EXPERIMENTS

### 3.1 SETUP

Training Data. Our training data is derived from the `alpaca_evol_instruct_70k`[2] dataset, a general instruction-following corpus. We performed a thorough cleaning and filtering process based on the following criteria: (1) Removed questions with excessively long ground truth answers to maintain manageable response lengths. (2) Eliminated noise such as HTML tags, non-alphanumeric characters, and duplicate entries. (3) Applied frequency-based filtering to exclude rare or long-tail queries and

---

[2]https://huggingface.co/datasets/WizardLMTeam/WizardLM_evol_instruct_70k

Table 1: PASR vs. other baselines. Compared to the base model, PASR achieves an average performance improvement of +4.8 and +8.2 on the two models, respectively.

| Methods | Public | Math | | | Reasoning | | Knowledge | | Comp. | Gene. | Sum. | AVG |
|---|---|---|---|---|---|---|---|---|---|---|---|---|
| | | GSM8K | MATH | AIME24 | ARC | GPQA | Wino | CSQA | Drop | MMLU | Xsum | |
| **Qwen2.5-7B** | | | | | | | | | | | | |
| Vanilla | - | 88.8 | 68.4 | 6.7 | 85.3 | 25.6 | 64.7 | 62.8 | 78.6 | 46.0 | 31.6 | 55.9 |
| Self-Refine+(Madaan et al., 2023) | NIPS'23 | 89.6 | 69.4 | 6.7 | 89.0 | 27.7 | 73.8 | 67.5 | 80.2 | 63.0 | 56.2 | 62.3 |
| Self-Refine(Shinn et al., 2023) | NIPS'23 | 88.7 | 68.4 | 16.7 | 85.3 | 25.6 | 64.1 | 62.3 | 78.6 | 49.0 | 36.0 | 57.5 |
| PTR(Du et al., 2025) | ICLR'25 | 88.6 | 61.8 | 10.0 | 91.0 | 27.7 | 59.0 | 75.3 | 75.7 | 74.0 | 50.4 | 61.6 |
| SCoRe(Kumar et al., 2025) | ICLR'25 | 82.4 | 63.2 | 3.3 | 67.2 | 14.5 | 48.1 | 46.4 | 65.8 | 56.0 | 35.0 | 48.2 |
| STaR(Zelikman et al., 2022) | NIPS'22 | 83.5 | 70.8 | 10.0 | 88.3 | 19.3 | 53.7 | 19.4 | 72.2 | 47.0 | 32.9 | 49.7 |
| ISC(Han et al., 2024) | AAAI'24 | 56.2 | 56.6 | 6.7 | 67.6 | 19.4 | 56.3 | 50.1 | 57.8 | 35.0 | 31.5 | 43.7 |
| RISE(Qu et al., 2024) | NIPS'24 | 84.9 | 62.4 | 13.3 | 82.9 | 23.7 | 60.9 | 74.5 | 73.1 | 45.0 | 56.6 | 57.7 |
| PASR(+prompt) | - | 79.0 | 54.4 | 6.7 | 46.8 | 22.5 | 34.8 | 30.3 | 70.6 | 34.0 | 23.1 | 40.2 |
| PASR(+IFT) | - | 89.2 | 70.8 | 3.3 | 84.6 | 23.6 | 62.4 | 65.4 | 77.3 | 51.0 | 42.0 | 57.0 |
| **PASR†** | - | **88.8** | **73.6** | 10.0 | 86.6 | **29.3** | 57.0 | 67.0 | **79.6** | **75.0** | 49.9 | **61.7** |
| **Qwen3-8B** | | | | | | | | | | | | |
| Vanilla | - | 91.3 | 80.2 | 13.3 | 89.0 | 25.0 | 64.5 | 66.3 | 71.2 | 72.0 | 36.3 | 60.9 |
| Self-Refine+(Madaan et al., 2023) | NIPS'23 | 94.8 | 84.4 | 23.3 | 94.0 | 43.7 | 83.0 | 83.5 | 85.0 | 85.0 | 51.1 | 72.8 |
| Self-Refine(Shinn et al., 2023) | NIPS'23 | 90.5 | 73.0 | 10.0 | 91.3 | 29.1 | 76.8 | 75.8 | 80.8 | 73.0 | 50.2 | 65.0 |
| PTR(Du et al., 2025) | ICLR'25 | 88.7 | 72.0 | 6.7 | 80.9 | 32.3 | 66.1 | 46.4 | 65.5 | 53.0 | 33.7 | 54.5 |
| SCoRe(Kumar et al., 2025) | ICLR'25 | 91.4 | 81.2 | 13.3 | 87.3 | 36.7 | 70.7 | 63.9 | 78.9 | 72.0 | 45.0 | 64.0 |
| STaR(Zelikman et al., 2022) | NIPS'22 | 72.7 | 55.2 | 0.0 | 64.2 | 26.0 | 55.3 | 28.8 | 49.5 | 22.0 | 13.7 | 38.7 |
| ISC(Han et al., 2024) | AAAI'24 | 23.6 | 57.2 | 6.7 | 68.2 | 29.2 | 63.5 | 28.3 | 42.5 | 28.0 | 38.3 | 38.6 |
| RISE(Qu et al., 2024) | NIPS'24 | 92.5 | 77.4 | 16.7 | 88.3 | 33.3 | 70.8 | 37.2 | 82.4 | 44.0 | 49.3 | 59.2 |
| PASR(+prompt) | - | 60.3 | 67.8 | 10.0 | 57.9 | 29.4 | 60.4 | 74.3 | 75.1 | 52.0 | 26.6 | 51.4 |
| PASR(+IFT) | - | 91.7 | 74.6 | 6.7 | 73.6 | 35.1 | 68.7 | 29.3 | 73.5 | 36.0 | 36.3 | 52.6 |
| **PASR†** | - | **94.9** | **81.4** | **16.7** | **92.3** | 24.5 | **80.0** | **79.6** | **85.3** | **83.0** | **53.0** | **69.1** |

low-frequency phrases that are unlikely to contribute effectively to the model's refinement capabilities. After these preprocessing steps, we obtained approximately 40,000 high-quality, open-domain query-answer pairs for training. We have release the training data in the GitHub.

**Benchmarks and Metrics.** We evaluate generalization of PASR across ten datasets covering diverse tasks. For general knowledge evaluation, we use MMLU (Hendrycks et al., 2021a). DROP (Dua et al., 2019) is included to assess multi-hop and comprehensive reasoning. Mathematical reasoning is evaluated using GSM8K (Cobbe et al., 2021), MATH (Hendrycks et al., 2021b), and AIME24 [3]. To test complex reasoning abilities, we adapt ARC [4] and GPQA [5]. Winogrande (Wino) (Sakaguchi et al., 2021) and CommonsenseQA (CSQA) (Talmor et al., 2019) are used for knowledge-based reasoning. For summarization, we use XSum dataset [6]. Accuracy is used as the evaluation metric for all datasets except XSum, for which we report similarity scores.

**Baselines.** We use Qwen2.5-7B (Qwen et al., 2025) and Qwen3-8B[7] as the backbone models, and compare PASR against several existing methods designed to induce self-improvement or self-correction abilities in LLMs. The baselines include: (1) **Self-refine** (Shinn et al., 2023): Prompts a base model to critique and iteratively revise its own responses in a single-turn format. (2) **Self-refine+** *(with oracle)* (Madaan et al., 2023): An enhanced version of Self-Refine that uses ground-truth answers as an oracle signal. The model first generates an initial response, then identifies errors by comparing with the ground truth and revises them accordingly. (3) **PTR** (Du et al., 2025): Constructs a progressive self-refinement dataset and applies instruction tuning to enable multi-turn, answer-level refinement. (4) **SCoRe** (Kumar et al., 2025): Employs a multi-turn reinforcement learning framework to train LLMs to self-correct without relying on oracle feedback. (5) **STaR** (Zelikman et al., 2022): Uses few-shot prompting to generate rationales for multiple questions. If the answer is incorrect, the rationale is regenerated using the correct answer. The model is iteratively fine-tuned on rationales that lead to correct outcomes. (6) **ISC** (Han et al., 2024): Builds a self-correction dataset and applies instruction tuning to train the model's intrinsic self-correction ability to detect and amend its own errors. (7) **RISE** (Qu et al., 2024): Creates improvement trajectories showing how a model can refine its own responses under its own distribution, and fine-tunes the model on these recursive rollouts.

---

[3] https://huggingface.co/datasets/math-ai/aime24

[4] https://huggingface.co/datasets/allenai/ai2_arc

[5] https://huggingface.co/datasets/Idavidrein/gpqa

[6] https://huggingface.co/datasets/EdinburghNLP/xsum

[7] https://huggingface.co/Qwen/Qwen3-8B

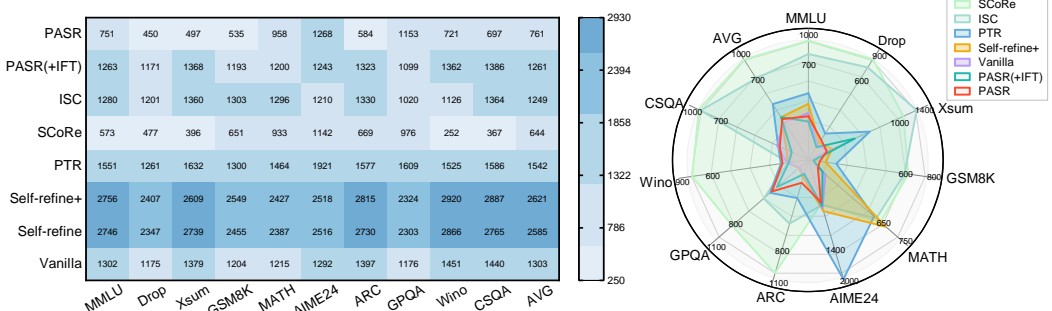

Figure 3: Comparison of average token length across different methods on various tasks. The left figure uses the Qwen3-8B backbone, while the right figure uses Qwen2.5-7B.

Detailed descriptions of the prompts, important parameters and implementation settings for all baselines are shown in the Appendix A.

## 3.2 Main Results

### 3.2.1 Performance Analysis of PASR

Unlike prior approaches that perform refinement only after the generation is complete, PASR refines answers adaptively during the generation process. To evaluate its effectiveness, we conduct experiments across a diverse set of tasks, with a focus on generalization capability. For fair comparison, we re-implement representative baselines that are only trained on specific domains under the same training data. The results are shown in Table 1.

**PASR consistently outperforms baseline models, with particularly notable gains on more challenging tasks.** For example, on the Qwen2.5-7B model evaluated with the MATH dataset, PASR yields a 5.2 improvement in accuracy compared to the standard method. Similarly, on the Qwen3-8B model tested with the Drop dataset, PASR achieves a 14.1 accuracy gain over the standard method. These results suggest that PASR, is capable of dynamically detecting and correcting reasoning errors, leading to effective and domain-agnostic performance gains.

**PASR achieves high performance without relying on external feedback or task-specific supervision.** We find that only when oracle feedback is available to assist refinement, the self-refine[+] provides the performance boost. This highlights the limitation of the self-refine structure in effectively improving model performance without external guidance , which is also observed in (Kumar et al., 2025; Qu et al., 2024). However, external supervision signals are often difficult to obtain and introduce additional costs. In contrast, PASR performs self-refinement autonomously, relying solely on intrinsic, self-adaptive decisions made during the generation process.

**PASR demonstrates strong generalization capabilities.** PASR is trained on general tasks and evaluated on domain-specific datasets to assess its generalization ability. Despite this domain shift, PASR achieves the best average performance compared to other self-refinement methods. While PASR does not always outperform all baselines on every individual dataset. For instance, its performance on Qwen2.5-7B is slightly lower on certain domain-specific tasks. This outcome is expected and understandable. Domain-specific tasks often require specialized knowledge or exhibit distributional characteristics not present in the training data. Moreover, we observe that the effectiveness of PASR can also vary with the underlying model. Compared to the more advanced Qwen3-8B, Qwen2.5-7B appears to exhibit a relatively weaker ability to leverage the learned proactive self-refinement mechanism. This suggests that stronger base models provide are fundamental to proactive self-refinement capability.

### 3.2.2 Efficiency Analysis of PASR

**PASR optimizes the output quality with minimal additional token overhead.** We compare token consumption across different baselines, as illustrated in Figure 3. Compared to standard decoding method, PASR achieves notable accuracy gains with only a slight increase in token usage. This highlights its ability to enhance outputs through targeted, dynamic refinements rather than full

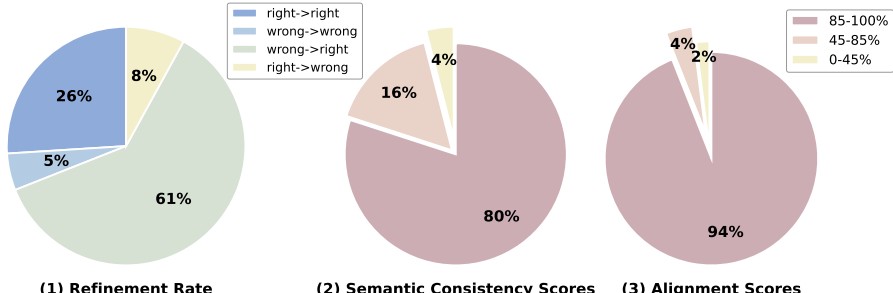

Figure 4: From left to right, the pie charts show: (1) the proportion of answers changed by PASR refinement, (2) the distribution of coherence scores reflecting how well the self-refinement builds upon the initial generation, and, and (3) the distribution of alignment scores measuring the consistency between the refinement process and the final answer. For (2) and (3), each segment represents the proportion of examples falling within a specific score range (e.g., [0–0.45), [0.45–0.85), [0.85–1.0]).

rewrites, making it a cost-efficient refinement method. Specifically, on the Qwen2.5-7B, PASR yields a 4.8 absolute performance improvement with only an 8.4% increase in token consumption compared to standard generation.

Additionally, while PASR and PTR achieve comparable performance on Qwen2.5-7B, PTR incurs significantly higher token costs. The performance gain of PTR mainly stems from the use of high-quality, answer-level refinement data. However, the effectiveness of this data diminishes considerably on Qwen3-8B. However, PTR regenerates entire answers at each refinement step, resulting in substantial token overhead.

### 3.3 DOES PASR GENUINELY EXHIBIT PROACTIVE REFINEMENT CAPABILITIES DURING GENERATION?

We investigate whether PASR performs proactive refinement during the generation process rather than passively correcting outputs after completion. To validate this, we conduct a quantitative analysis from three complementary perspectives: (1) whether PASR performs refinement at appropriate moments; (2) whether the refinement behavior modifies earlier reasoning steps or simply regenerates content; (3) whether these refinements contribute causally to improving the final output quality. The prompts used in this subsection are shown in Table 7 and 8. The results are shown in the Figure 4.

**PASR autonomously determine when to refine.** We randomly sample 384 questions, among which 267 are initially answered incorrectly by the base model. PASR does not refine all answers indiscriminately; instead, it selectively triggers refinement. Among the 267 incorrect answers, 235 are revised and corrected by PASR. While many originally correct answers nearly remain unchanged. This indicates that PASR is able to identify and act upon potentially flawed generations when refinement is necessary.

**PASR shows high coherence between pre- and post-refinement outputs.** We randomly sample 300 answers and employ an independent LLM, Qwen2.5-32B-Instruct, to evaluate their semantic consistency before and after refinement. Each sample is scored multiple times within in $[0, 1]$ to ensure the reliability of the assessment. The results indicate that nearly 80% of samples received a semantic consistency score exceeding 0.9.

**PASR's proactive self-refinement process contributes to the answer correctness.** We further analyze the 300 samples mentioned above to evaluate the alignment between the refinement process and the final answer. Over 85% of the samples achieved a alignment score above 0.9, indicating that refinement leads to the quality of outputs.

### 3.4 WHAT MAKES PASR EFFECTIVE?

**Reinforcement learning enables the model to perform proactive self-refinement.** In contrast, *prompt-based or supervised signals are insufficient to elicit proactive refinement capabilities.* We explore whether proactive self-refinement can be induced via prompting. The results are shown in

Table 1. When the model is explicitly instructed to self-refine during generation via prompt design (PASR+prompt), we observe a *consistent performance decline* across all tasks, with an average decrease of 16.9 and 9.5 on two backbone models. It indicates that prompt-based guidance alone is insufficient to elicit the model's adaptive self-refinement capability.

Similarly, we apply instruction-following finetuning (PASR+IFT) to inject this capability. However, the model shows *limited generalization* to unseen tasks. On the Qwen3-8B model, performance drops by 8.3 compared to the base version. These results suggest that proactive self-refinement is not an innate capability and cannot be effectively acquired through SFT.

**Comparison-based rewards setting help the model learn to perform effective refinements.** We use Qwen2.5-7B as the backbone and evaluate the effectiveness of two alternative reward strategies. The first is *Single-reference comparison (w/o multi-answer)*, computes refinement rewards by comparing the refined output to *a single standard answer*. The second is *Refinement-triggered reward (w/o comparison)*, assigns a coarse positive refinement reward whenever a refinement action is taken, regardless of its necessity or effectiveness. The results are shown in Table 2. This reward strategy offers several key advantages.

First, averaging over multiple standard answers reduces the variance introduced by the randomness of LLM outputs. It provides a more *robust and stable* learning signal for guiding meaning-

Table 2: PASR performance across datasets under different refinement reward signals. The comparison-based fine-grained reward better guides the model to learn adaptive and meaningful refinements.

| Dataset | PASR | w/o multi-answer | w/o comparison |
|---------|------|------------------|----------------|
| MMLU | 75.0 | 71.0 (-4.0) | 53.0 (-22.0) |
| Drop | 79.6 | 76.7 (-2.9) | 78.6 (-1.0) |
| Xsum | 49.9 | 44.3 (-5.6) | 31.9 (-18.0) |
| GSM8K | 88.8 | 75.7 (-13.1) | 86.0 (-2.8) |
| MATH | 73.6 | 62.2 (-11.4) | 62.2 (-11.4) |
| AIME24 | 10.0 | 10.0 (+0.0) | 10.0 (+0.0) |
| ARC | 86.6 | 83.9 (-2.7) | 82.9 (-3.7) |
| GPQA | 29.3 | 28.9 (-0.4) | 27.4 (-1.9) |
| Wino | 57.0 | 53.4 (-3.6) | 65.3 (+8.3) |
| CSQA | 67.0 | 65.9 (-1.1) | 64.9 (-2.1) |
| **AVG** | **61.7** | 57.2 (-4.5) | 56.2 (-5.5) |

ful refinements during training. This strategy enables the model to better recognize when a refinement yields a genuine improvement. Moreover, coarse-grained reward signals are easily exploited by the model, leading to unnecessary refinement in pursuit of high reward (i.e., *reward hacking*). In contrast, our comparison-based signal avoids this by rewarding only measurable improvements, leading to more targeted and meaningful refinements.

## 4 RELATED WORK

**Prompt-based self-refinement.** Prior work on self-refinement typically follows a two-stage paradigm. The model first generates an initial response and is then prompted to refine or improve it (Ganguli et al., 2023a). These methods have seen widespread use in complex reasoning tasks, including math (Weng et al., 2023; Wang et al., 2024) and code generation (Olausson et al., 2023b; 2024; 2023a). However, simply prompting a model to refine its own output does not consistently yield better results, and there is little evidence that prompting alone is sufficient for reliable self-improvement(Huang et al., 2024; Tyen et al., 2024). Success in these settings often relies on the availability of ground truth feedback or external supervision, such as explicit information about the error, its location, and an explanation of why it is wrong (Kim et al., 2023; Shinn et al., 2023). Unfortunately, such fine-grained feedback is rarely accessible in practical applications (Gou et al., 2024; Pan et al., 2024). Therefore, some studies utilize stronger models or train auxiliary teacher models to evaluate outputs and provide feedback (Xie et al., 2025; Madaan et al., 2023; Uesato et al., 2023; Welleck et al., 2023). While effective, these approaches usually require task-specific annotations to train the feedback models, which significantly increases the cost and limits scalability across diverse tasks (Du et al., 2025).

**Fine-tuning for self-refinement.** Another line of work focuses on SFT using synthetic self-refinement data. In these settings, initial answers are generated by one model, while refined answers are produced by a stronger model or taken from oracle answers (Havrilla et al., 2024; Du et al., 2025; Han et al., 2024) (Xie et al., 2025). The resulting pairs of "bad" to "good" answers are used to train models to imitate the refinement process. However, such methods suffer from either distributional mismatch, where the errors in training data do not reflect the mistakes the model makes during inference (Kang

et al., 2025), or behavioral collapse, where the model learns a narrow correction pattern that fails to generalize across tasks or domains (Kumar et al., 2025; Qu et al., 2024).

## 5 CONCLUSION

We propose PASR, a novel method that enables large language models to proactively self-refine their responses during generation. PASR leverages an on-policy reinforcement learning approach to explore whether, when, and how to perform refinements. We design fine-grained rewards to encourage effective refinements and penalize incorrect or unnecessary ones. Experiments show that PASR achieves a strong balance between performance and efficiency. Moreover, even when trained only on general open-domain data, PASR achieves strong self-refinement across ten diverse tasks, demonstrating strong generalization not observed in previous work.

## 6 ACKNOWLEDGMENTS

This work was supported by Ant Group and the ECNU Academic Innovation Promotion Program for Excellent Doctoral Students (Grant No. YBNLTS2024-007).

## 7 ETHICS STATEMENT

This work adheres to the ICLR Code of Ethics. In this study, no human subjects or animal experimentation was involved. All datasets used were sourced in compliance with relevant usage guidelines, ensuring no violation of privacy. We have taken care to avoid any biases or discriminatory outcomes in our research process. No personally identifiable information was used, and no experiments were conducted that could raise privacy or security concerns. We are committed to maintaining transparency and integrity throughout the research process.

## 8 REPRODUCIBILITY STATEMENT

We have made every effort to ensure that the results presented in this paper are reproducible. All code and datasets have been made publicly available in the github repository to facilitate replication and verification. The experimental setup, including training steps, model configurations, and hardware details, is described in detail in the paper. Additionally, All datasets are publicly available, ensuring consistent and reproducible evaluation results. We believe these measures will enable other researchers to reproduce our work and further advance the field.

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

# APPENDIX

## A EXPERIMENTAL DETAILS

### A.1 IMPLEMENTATION DETAILS FOR PASR

**Platform.** All of our experiments are conducted on workstations equipped with eight NVIDIA A800 PCIe GPUs with 80GB memory, running Ubuntu 20.04.6 LTS and PyTorch 2.0.1. About the training cost, using Qwen2.5-7B as an example, we train PASR with the following setup: 2 GPUs for rollout generation, 1 GPU for policy updates, and 1 GPU for hosting the reference model server. Training for 3,000 steps takes approximately 8 hours in total.

**Important Parameters of PASR.** The PASR is implemented based on the open-source GitHub repository [8]. The KL divergence penalty coefficient $\beta$ is set to 0.04 to balance policy improvement and deviation from the reference policy. The clipping parameter $\epsilon$ is set to 0.2. For each group, 8 answers are generated, and the training batch size is set to 2.

Distributed training utilizes the DeepSpeed library with the *AdamW* optimizer and a learning rate of 1e-6. Gradient accumulation occurs over 4 steps, and with a per-GPU batch size of 2, the effective batch size is $8 \times N_{\text{GPUs}}$, where $N_{\text{GPUs}}$ denotes the number of GPUs.

Mixed-precision training with BF16 is enabled. Memory optimization employs ZeRO Stage 2, with optimizer state offloading to CPU. Key ZeRO configurations include allgather partitions, an allgather bucket size of 2e8, reduce scatter, and a reduce bucket size of 2e8. Contiguous gradients are enabled, communication overlap is disabled, and 16-bit weights are gathered during model saving. Training loss is logged every 5 steps.

**Details on the Judge Model.** During training, we employed Qwen2.5-32B-Instruct as the judge model, which has been widely adopted for assessing answer correctness (Yu et al., 2025). To ensure reliable and objective evaluation, our prompt design explicitly incorporated three elements: the question, the ground truth, and the model-generated answer. The judge model was instructed to ground its judgment on the provided ground truth rather than on subjective impressions, thereby avoiding inconsistent criteria and yielding more stable evaluations than direct answer-only comparisons. The full evaluation prompts used in both training and testing are shown in Table 6.

To verify the trustworthiness of the judge model, we randomly sampled 50 evaluation cases from the test set and performed manual verification. Each case was independently reviewed by two human annotators, who compared the generated answer against the ground truth. We observed a 91% agreement rate between the judge model's assessments and human judgments, confirming that the judge model provides consistent and reliable scoring.

For deployment, the judge model runs on four A800 (80GB) GPUs with a batch size of 8, achieving an evaluation speed of approximately 43.27 tokens per second (about 2 seconds per batch).

Table 3: PASR vs. other baselines. Compared to the base model, PASR achieves an average performance improvement of 4.9% on Qwen2.5-14B.

| Methods | Public | Math | | | Reasoning | | Knowledge | | Comp. | Gene. | Sum. | AVG |
|---|---|---|---|---|---|---|---|---|---|---|---|---|
| | | GSM8K | MATH | AIME24 | ARC | GPQA | Wino | CSQA | Drop | MMLU | Xsum | |
| **Qwen2.5-14B** | | | | | | | | | | | | |
| Vanilla | - | 92.9 | 75.6 | 20.0 | 89.0 | 38.4 | 81.1 | 66.4 | 87.5 | 57.0 | 60.5 | 66.8 |
| Self-Refine[+](Madaan et al., 2023) | NIPS'23 | 93.6 | 78.0 | 30.0 | 92.3 | 46.3 | 88.1 | 74.0 | 92.3 | 73.0 | 57.1 | 72.5 |
| Self-Refine(Shinn et al., 2023) | NIPS'23 | 92.3 | 75.2 | 20.0 | 89.0 | 38.5 | 80.2 | 65.7 | 86.9 | 57.0 | 57.2 | 66.2 |
| PTR(Du et al., 2025) | ICLR'25 | 87.6 | 63.6 | 10.0 | 86.6 | 37.0 | 84.5 | 75.3 | 83.7 | 54.0 | 44.3 | 62.7 |
| SCoRe(Kumar et al., 2025) | ICLR'25 | 93.3 | 78.2 | 10.0 | 86.3 | 44.1 | 86.8 | 70.5 | 84.6 | 80.0 | 70.9 | 70.5 |
| STaR(Zelikman et al., 2022) | NIPS'22 | 87.0 | 75.4 | 6.7 | 87.0 | 39.2 | 78.0 | 70.2 | 89.5 | 72.0 | 63.2 | 66.8 |
| ISC(Han et al., 2024) | AAAI'24 | 88.1 | 64.0 | 23.3 | 77.9 | 35.2 | 71.2 | 62.9 | 83.7 | 75.0 | 46.2 | 62.8 |
| PASR(+prompt) | - | 88.7 | 71.6 | 26.7 | 78.9 | 26.3 | 71.0 | 68.0 | 88.5 | 66.0 | 17.7 | 60.3 |
| PASR(+IFT) | - | 75.0 | 59.4 | 23.3 | 86.0 | 38.4 | 67.4 | 69.0 | 78.9 | 68.0 | 61.3 | 62.7 |
| **PASR†** | - | 93.6 | 78.0 | 30.0 | 88.8 | 45.1 | 86.0 | 78.3 | 89.9 | 74.0 | 53.2 | 71.7 |

---

[8] https://github.com/lsdefine/simple_GRPO

## A.2 IMPLEMENTATION DETAILS FOR BASELINES

We use the LLaMA-Factory framework[9] to train all baseline methods. The key parameters are shown in the Table 4.

Table 4: Important parameters for each baseline method

| Method | Parameters |
| --- | --- |
| **PTR** | per_device_train_batch_size: 1
gradient_accumulation_steps: 2
learning_rate: $1.0 \times 10^{-5}$
num_train_epochs: 2
lr_scheduler_type: cosine
warmup_ratio: 0.1
bf16: true
Dataset: Public GitHub |
| **SCoRe** | per_device_train_batch_size: 1
gradient_accumulation_steps: 4
learning_rate: $1.0 \times 10^{-5}$
num_train_epochs: 2.0
lr_scheduler_type: cosine
warmup_ratio: 0.1
bf16: true
Dataset: preference pairs form PTR experiment |
| **STaR** | per_device_train_batch_size: 1
gradient_accumulation_steps: 2
learning_rate: $1.0 \times 10^{-5}$
num_train_epochs: 2
lr_scheduler_type: cosine
warmup_ratio: 0.1
bf16: true
Dataset: alpaca_evol_instruct_70k(filtered generated pairs)) |
| **ISC** | per_device_train_batch_size: 1
gradient_accumulation_steps: 2
learning_rate: $1.0 \times 10^{-5}$
num_train_epochs: 2.0
lr_scheduler_type: cosine
warmup_ratio: 0.1
bf16: true
Dataset: alpaca_evol_instruct_70k |
| **RISE** | per_device_train_batch_size: 1
gradient_accumulation_steps: 2
learning_rate: $1.0 \times 10^{-5}$
num_train_epochs: 2.0
lr_scheduler_type: cosine
warmup_ratio: 0.1
bf16: True
Dataset: alpaca_evol_instruct_70k |
| **PASR(+IFT)** | per_device_train_batch_size: 1
gradient_accumulation_steps: 2
learning_rate: $1.0 \times 10^{-5}$
num_train_epochs: 2.0
lr_scheduler_type: cosine
warmup_ratio: 0.1
bf16: True
Dataset: good refinement paths generated during PASR training |

---

[9] https://github.com/hiyouga/LLaMA-Factory

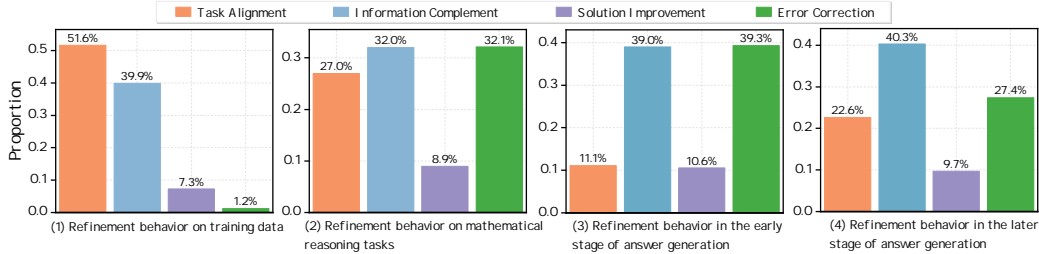

Figure 5: The frequency distribution of the four refinement types in PASR.

# B FURTHER ANALYSIS

## B.1 FURTHER PERFORMANCE ANALYSIS OF PASR

As shown in Table 1, PASR improves the average performance of Qwen2.5-7B and Qwen3-8B by 4.8 and 8.2 points, respectively, compared to standard generation across ten benchmarks. We further evaluate PASR on Qwen2.5-14B and DeepSeek-R1-Distill-Llama-8B[10], the results are reported in Table 3.

Notably, PASR yields larger gains on models with stronger reasoning capabilities. For example, it boosts the average accuracy of Qwen3-8B by 8.2 points, whereas the improvement on the larger Qwen2.5-14B model is 4.9 points. This pattern suggests that PASR's effectiveness does not simply scale with model size. Instead, it demonstrates PASR's intrinsic reasoning ability to generalize across different model families, delivering consistent improvements regardless of model configuration.

## B.2 REFINEMENT BEHAVIOR ANALYSIS OF PASR

This experiment aims to investigate how PASR autonomously refines its outputs during generation, including the types of refinement behaviors it exhibits and the factors that limit its effectiveness. Specifically, we analyze both qualitative examples and quantitative statistics of refinement types, and examine failure cases to understand the model's strengths and inherent constraints.

**Refinement behavior examples of PASR.** In the Section 2, we define four intended refinement behaviors of PASR, including Error Correction, Information Complement, Solution Improvement, and Task Alignment. While these four categories guide the design of the system prompt during training, PASR is not explicitly instructed to follow a specific type when solving tasks. Instead, the model autonomously decides the appropriate refinement behavior based on the task context. We provide a concrete example for each of the four refinement types to clearly demonstrate how PASR operates. Examples are shown in Table 5.

**Statistical analysis of the four refinement types.** We sample 2,678 refinement outputs from PASR's training process and used GPT-4o to classify the type of refinement performed. The prompt used is shown in Table 6 and the results are shown in Figure 5(1). We find that PASR mainly performs *Task Alignment and Information Complement*. This pattern is related to the training data, which consists mostly of general instruction-tuning corpora. As a result, the model tends to ensure task compliance and complete missing information during generation, rather than focus on structural changes or post-hoc error correction.

Morever, to validate PASR's refinement behavior in reasoning tasks, we randomly sampled 300 answers from the math benchmark and analyzed the distribution of the four refinement types. The results are show in Figure 5 (2). These results indicate that PASR's behavior distribution varies across task types: Error Correction and Information Complement are predominant in mathematical reasoning tasks, whereas Task Alignment and Information Complement are more common in open-ended tasks.

We further investigated how PASR's refinement behavior evolves during the generation process by dividing each answer into two halves. The results are show in Figure 5 (3) and (4). They show that Error Correction and Information Complement remain the most frequent behaviors throughout the generation. Notably, Information Complement becomes more prominent in the later stage, suggesting

---

[10]https://huggingface.co/deepseek-ai/DeepSeek-R1-Distill-Llama-8B

that PASR tends to add supplementary information toward the end of reasoning to enhance output quality.

**Error Case Analysis.** We conducted an analysis of PASR's failure cases to better understand its limitations. As discussed in Section 3.2.2. Among 267 questions initially answered incorrectly, PASR successfully corrected 235 through refinement, while 32 questions remained incorrect. Manual inspection of these 32 cases revealed two main reasons for failure. First, questions beyond knowledge boundaries. These involved the question outside the model's existing knowledge, and self-refinement cannot introduce new information, similar to the limitations of human self-correction. This represents an inherent limitation of current models rather than a shortcoming of PASR, and identifying such cases can guide future targeted improvements. Second, limited metacognitive ability of existing LLMs. The model sometimes fails to accurately recognize or locate its own errors. This restricts the refinement process, causing it to only partially address or overlook core mistakes.

**Summary.** PASR's refinement behavior is inherently adaptive: the model chooses how and when to refine outputs based on the specific task, demonstrating the advantage of autonomous, context-aware refinement. However, self-refinement has intrinsic limitations. It cannot fully correct errors that stem from gaps in the model's knowledge or limitations in its reasoning capabilities.

### B.3 DISCUSSION ON HOW PASR MITIGATES OVER-THINKING.

Over-thinking in LLMs often results from redundant or unfocused thought processes, leading to unnecessarily long outputs. PASR alleviates this issue by enabling the model to perform more targeted and effective self-refinement, which naturally produces shorter and more purposeful responses. Empirical results support this effect. As shown in Figure 3, PASR generates consistently shorter outputs compared to other self-refinement methods. This behavior is further encouraged by our reward function, which reinforces effective refinements, penalizes detrimental ones, and discourages unnecessary refinement actions. Notably, even without explicitly penalizing output length, PASR achieves more concise reasoning by focusing on meaningful refinements, demonstrating that enhancing the quality and efficiency of self-refinement can reduce over-thinking.

---

**Four Refinement Types of PASR**

**/\* Case1: Exmaple of the Error Correction behavior in PASR \*/**

Question:

In the following sentences, what are the commonly confused words that can be corrected using Latex and what is the difference between them?

Answer:

**\<think\>**First, I need to identify the commonly confused words and then I'll explain the differences between them using LaTeX. After that, I'll provide multiple examples for each word to show their correct usage. **\<refine\>Oops! I forgot to specify the words in LaTeX format. Let me fix that.\</refine\>**

*( ... more reasoning tokens omitted for brevity )*

**\</think\>**

**\<answer\>**These explanations and examples illustrate the differences between the commonly confused words and demonstrate their correct usage. The words and their differences are clearly indicated using LaTeX, and multiple examples prove the model's understanding and application of the correct usage in context. **\</answer\>**

**/\* Case2: Exmaple of the Iormation Complement beha behavior in PASR \*/**

Question:

---

Can you write a Python script that scrapes data from PubMed related to the relationship between the endocannabinoid system and the immune system?.
Answer:
**<think>**

To complete this task, I will need to use Python's requests and BeautifulSoup libraries to scrape the data from PubMed.

*( ... more reasoning tokens omitted for brevity )*

Finally, I need to add the necessary error handling to make sure that the script can handle any errors that might occur during the scraping and parsing process.
**<refine>I should also add some code to parse the abstract of each article to extract any relevant information related to the endocannabinoid system and the immune system. This will help to further analyze the current state of research on this topic. </refine>**
**</think>**
**<answer>**
Here is the Python code that I have written to complete the task: *( ... )*

Any articles that were not published within the last five years are filtered out. The data is then stored in a CSV file using the csv library.

**</answer>**

**/\*Case 3: Exmaple of the Solution Improvement beha behavior in PASR \*/**
Question:
Can you provide an overview of the labor movement using a historical timeline? ...
Answer:
**<think>**

My initial thought is that we can start by researching the key events in the labor movement's history.
*( ... more reasoning tokens omitted for brevity )* To make it more engaging, I should include

hyperlinks for each event that would lead to a brief description.

**<refine>Suggestion to include hyperlinks is a good idea. This would make the timeline more interactive. However, we should also consider adding images or embedded videos relevant to the events for a more visual representation. Let's refine this in our approach. </refine>**
**</think>**
**<answer>**
*( ... )*

**</answer>**

**/\*Case4: Exmaple of the Task Alignment beha behavior in PASR \*/**

Question:

Utilizing advanced sentiment analysis in MATLAB, can you determine the nuanced emotional tone conveyed in this quote and provide further insight into Atticus Finch's complex attitude towards the Chief Justice and the United States Constitution? ...

Answer:

**<think>**

*( ...  more reasoning tokens omitted for brevity )* Additionally, we should consider the

historical and political context of the quote to better understand Atticus Finch's complex attitude towards the Chief Justice and the United States Constitution

**<refine>The question asks us to delve deeper into the historical and political context of the novel, particularly in relation to the societal and legal injustices prevalent during that time period. How does the quote reflect these issues, and what can it tell us about the societal norms and legal system of the time? </refine>**

**</think>**

**<answer>**

*( ... )*

**</answer>**

---

**All Prompts Used in PASR**

**/\* Prompt Template for PASR \*/**

System:

You are a helpful assistant with self-refinement capability. After the user asks a question, you first think carefully and then give the answer.

The thinking process and answer should be enclosed within <think> </think> and <answer> </answer> tags respectively. Note that you can only use once these four tags.

In the <think> and </think> tag, follow these rules:

Start with an initial thought process on how to approach the question.

**when you determine that additional clarification, detail, or improved reasoning is necessary, insert <refine> </refine> tag and then specify what needs to be reconsidered or improved. You can use both tags multiple times.**

Continue to advance your reasoning after each refinement until you feel there is no more room for improvement.

This is how your full response should be structured:

<think>Here is your thinking process, when you think you need to reflect, insert <refine>your refinement</refine>. Repeat the iterative process as many times as necessary before moving

to the final answer.</think><answer>Here is an answer at the end of the thinking process.</answer>

**/\* Prompt Template for ASR evaluation \*/**

You are a judger, you will judge the correctness of the answer to the question. Below is a question, a ground truth answer, and an answer generated by an AI assistant, please rate the AI assistant's answers according to the question on a scale from 0 to 1. Your output is just a number in the range from 0 to 1.

### Question:
{Question}

### Ground Truth:
{Ground Truth}

### Answer:
{Answer}

**/\* Prompt for Evaluating the Refinement Behavior of PASR \*/**

You are a judge of content refinements. I will give you a sentence that may contain one or more <refine>...</refine> sections. Each <refine> section shows a refinement made by a model.

Step 1: Analyze each <refine> section one by one.

- For each <refine>, classify it into exactly one of the following types:

1. Error Correction: fixes factual, logical, or calculation mistakes.

2. Information Complement: adds missing important details to make content complete and correct.

3. Solution Improvement: improves the effectiveness or efficiency of the solution.

4. Task Alignment: adjusts the content to better match the task goal or user intent.

- Output only the category name for each <refine>, in order, one per line.

Step 2: After all refinements are labeled, create a JSON summary of counts for each type.

Example format:

{
"Error Correction": 0,
"Information Complement": 0,
"Solution Improvement": 0,
"Task Alignment": 0
}

Rules:

- Do not add any explanations, only output the category names first, then the final JSON summary.

- Use the exact category names above.

Here is the sentence: {sentence_with_refinement}

**/* Evaluation Prompt Template for Summary Questions */**

Now, I want to test an AI assistant's ability to summary. Below is a text (Question), a ground truth summary (Ground Truth Answer), and an answer (Answer) generated by an AI assistant. Please rate the AI assistant's answers according to the ground truth answer. Please score answers according to how relevant they are to the text and ground truth summary. Your output is from 0 to 1,which 0 is not similar at all, 1 is basically error free.

### Question:

Ground Truth:Ground Truth

Answer:Answer

**/* Evaluation Prompt Template for Multiple-Choice Questions */**

Now, I want to test an AI assistant's ability to answer questions. Below is a multi-choice question, a ground truth answer(one of the option), and an answer generated by an AI assistant. Please rate the AI assistant's answers according to the question and the ground truth answer. If you think the answer is correct, your output is 1; otherwise, your output is 0.Your output is just 0 or 1.

### Question:

Ground Truth:Ground Truth

Answer:Answer

**/* Evaluation Prompt Template Open Questions */**

Now, I want to test an AI assistant's ability to answer questions. Below is a open question, a ground truth answer, and an answer generated by an AI assistant. Please rate the AI assistant's answers according to the ground truth answer. If you think the answer is correct, your output is 1; otherwise, your output is 0. Your output is just 0 or 1.

Question:Question Ground Truth:Ground Truth Answer:Answer

**/* Prompt Template for Refinement with Oracle (Math Questions) */**

There might be an error in the solution above because of lack of understanding of the question. Please correct the error, if any, and rewrite the solution. Only output the final solution! At the end of the Solution, when you give your final answer, write it in the form Final Answer: The final answer is \box{answer}. I hope it is correct.

### previous solution:Initial answer

**/* Prompt Template for Refinement without Oracle (Open Questions) */**

There is an error in the previous solution. Please review each step to identify the mistake, and then provide a corrected version of the solution.

### previous solution:Initial answer

**/* Prompt Template for Refinement without Oracle */**

Please review each step of the previous solution to identify any potential errors. If you find any issues, provide a revised and corrected version of the solution. If there are no issues, simply respond with: I believe the above solution is correct.

### previous solution:Initial answer

/* **Standard Prompt for MMLU */**

Here is a multiple-choice question, which from a dataset tests knowledge across 57 diverse fields such as elementary mathematics, history, computer science, and law. please think step by step and give me your final answer.

/* **Standard Prompt for Drop */**

Here is a passage and a question, which requires discrete reasoning over the provided text. Please think step by step and give me your final answer.

/* **Standard Prompt for Xsum */**

Here is a passage. please summarize this passage.

/* **Standard Prompt Template for Math (GSM8K, MATH, AIME24) */**

Here is a problem. please think step by step and give me your final answer.

/* **Standard Prompt for ARC */**

Here is a multiple-choice question, which from a collection of questions for the science exam. Please think step by step and give me your final answer.

/* **Standard Prompt for Wino */**

Here is a question provides two options. Please think step by step and select the correct answer based on the semantics of the sentence.

/* **Standard Prompt for CommonsenseQA */**

Here is multiple-choice about commonsense. Please think step by step and give me your final answer.

---

**Prompt for Evaluating the Reasonableness of the Refinement Process**

# Role
You are an AI Analyzer specializing in assessing the quality of refinement thinking.
# Task
Your task is to evaluate the "reasonableness" of the refinement part within a given response. This response typically contains two parts: an initial thought or response (pre-refinement), and a part where the user reflects on that initial thought (post-refinement).
# Definition of "Reasonableness"
"Reasonableness" here has a specific meaning: it measures the **coherence and consistency between the pre-refinement and post-refinement thought processes.**
You need to determine:
1. Is the refinement **based on** the preceding thought content?
2. Does the refinement process **logically follow** from the previous thinking? Or, if the refinement leads to a **shift in perspective**, is this shift explained or internally logical and understandable?

3. Does the conclusion or state after refinement form an understandable and **coherent thought trajectory** with the pre-refinement state?

**Crucially:** You are **not** evaluating the depth of the refinement itself, nor the correctness of the final answer. You are evaluating **only** whether the **act of refinement** is **coherent and consistent** with the preceding thought content.

# Evaluation Criteria & Score

Please provide a floating-point score between **0.0 and 1.0** based on the following criteria:

* **0.0:** Completely unreasonable. The refinement is entirely unrelated to the previous thinking, or contradicts it without any explanation. The thought process is broken or disconnected.

* **0.5:** Partially reasonable. The refinement has some connection to the previous thinking, but the link is weak, the logical chain is unclear, or a shift in perspective seems somewhat abrupt but has a faintly traceable thread.

* **1.0:** Highly reasonable. The refinement is clearly built upon the previous thinking, the logic is coherent, and even if perspectives shift, the reasons and process are clear, demonstrating high consistency in the thought trajectory.

# Output Requirements

* **Strictly output only a single number**, which must be a floating-point number between 0.0 and 1.0.

* **Do not include any** explanations, justifications, text descriptions, units, or any other extra characters.

# Response Text to Evaluate

---

## Box 8: Prompt for Evaluating the Consistency between the Refinement and the Final Answer

# Role

You are an AI Analyzer specializing in evaluating thought coherence.

# Task

Your task is to evaluate the consistency between a given "Thought Process" (which may include refinement) and the final "Answer".

# Definition of "Consistency"

"Consistency" here measures: **The degree to which the final answer is a direct, relevant, and logical product of the thought process.**

You need to determine:

1. Does the final answer directly address or resolve the problems, dilemmas, or goals explored in the thought process?

2. Is the final answer logically aligned with the thought process, including insights or conclusions derived from refinement?

3. Are the key information, reasoning steps, or refinements from the thought process reflected or applied in the final answer?

**Focus:** You are **not** evaluating the quality of the thought process itself, nor the correctness or merit of the answer itself. You are evaluating **only the degree of relevance and logical connection between the thought process and its final answer.**

# Evaluation Criteria & Score

Please provide a floating-point score between **0.0 and 1.0** based on the following criteria:

* **0.0:** Completely inconsistent/irrelevant. The final answer has little to no relation to the thought process, appears out of nowhere, or completely ignores the reasoning path.

* **0.5:** Partially consistent/relevant. The final answer has some connection to the thought process, but might only address parts of it, the logical link might be weak, or the answer, while related, doesn't seem like the most direct conclusion from the process.

* **1.0:** Highly consistent/relevant. The final answer clearly, directly, and logically stems from the provided thought process, serving as its definite conclusion or solution.

# Output Requirements

* **Strictly output only a single number**, which must be a floating-point number between 0.0 and 1.0.

* **Do not include any** explanations, justifications, text descriptions, units, or any other extra characters.

# Response Text to Evaluate

<think> </think> is thinking process, <answer> </answer> is final answer.

## C   LLM USAGE

Large Language Models (LLMs) were used to aid in the writing and polishing of the manuscript. Specifically, we used an LLM to assist in refining the language, improving readability, and ensuring clarity in various sections of the paper. The model helped with tasks such as sentence rephrasing, grammar checking, and enhancing the overall flow of the text.

It is important to note that the LLM was not involved in the ideation, research methodology, or experimental design. All research concepts, ideas, and analyses were developed and conducted by the authors. The contributions of the LLM were solely focused on improving the linguistic quality of the paper, with no involvement in the scientific content or data analysis.

The authors take full responsibility for the content of the manuscript, including any text generated or polished by the LLM. We have ensured that the LLM-generated text adheres to ethical guidelines and does not contribute to plagiarism or scientific misconduct.

