# OpenReview forum: "A Stitch in Time Saves Nine: Proactive Self-Refinement for Language Models"
_ICLR.cc/2026/Conference — ICLR 2026 Poster_

### Official Review · Reviewer_zxHQ · 2025-10-31

**Soundness:** 3
**Presentation:** 4
**Contribution:** 4
**Rating:** 6
**Confidence:** 3

**Summary:**

This paper introduces ProActive Self-Refinement (PASR) as a novel method for enabling Large Language Models (LLMs) to refine their outputs during the generation process, rather than as a post-hoc step.

The authors formalize this as a MDP and use RL to train models to decide whether, when, and how to refine their reasoning trace.

Key contributions include:
* A formulation of proactive self-refinement and trained the model with on-policy RL with GRPO.
* The design of a fine-grained, comparison-based reward function that encourages meaningful refinements while penalizing unnecessary or harmful ones.
* Extensive experiments on 10 diverse tasks using Qwen models, demonstrating that PASR improves accuracy and problem-solving performance.

**Strengths:**

* The core idea of shifting self-refinement from a reactive, post-hoc process to a proactive, in-process one is a novel and important conceptual leap.
* The choice of formulating proactive refinement as a Markov Decision Process and using RL is appropriate for the task of learning a sequential decision-making policy.
* The hybrid reward design, which combines rule-based format rewards with model-based accuracy and refinement rewards, is thoughtful and comprehensive.
* The evaluation is extensive, spanning 10 different datasets that cover math, reasoning, knowledge, and generation.

**Weaknesses:**

* Confusing claim about the token consumption:
  * In page 1, Lines 023-024: the author mentioned "PASR reduces average token consumption by 41.6%"
  * In Page 8, LIne 384 - 386, the author mentioned "with only an 8.4% increase in token consumption compared to standard generation."
  * It would be good to have a clear claim and explanation about the token consumption. So far, the claim of reduce by 41.6% is a claim not well supported.

* The analysis in the Fig. 6 reveals that the model predominantly learns "Task Alignment" and "Information Complement" behaviors, with "Error Correction" being the least frequent type. The authors attribute this to the nature of the training data, but it may point to a limitation in the model's ability to identify and fix deep logical or factual fallacies. "Text Alignment" is a fragile section that there is concern that the approach is simply learning to hack the reward model by overfitting to its stylistic preference.

**Questions:**

* Question about the token consumption. How does the 41.6% calculated. What's the setup to achieve this.

* The performance is good but not outstanding, especially when compared with self-refine (row 2 in Table 1). In what types of scenarios does the additional complexity of RL training provides additional advantage.

* It will be good to have more analysis on the pattern of timing of refinement: The model learns when to trigger a refinement. Does the model tend to refine early in its reasoning process to correct its trajectory, or does it make more refinements later once it has generated more context? Understanding these patterns could provide valuable insights into the model's learned metacognitive abilities.

* nit: Page5, Ln 238, duplicated "Consequently"

---

> ### Author Response · Authors · 2025-11-20
> **Response to Reviewer zxHQ**
>
> Thank you for your detailed review. We have carefully considered your suggestions and provide our responses below.
> > **Q1:** Confusing claim about the token consumption:
> - In page 1, Lines 023-024: the author mentioned "PASR reduces average token consumption by 41.6%"
> - In Page 8, LIne 384 - 386, the author mentioned "with only an 8.4% increase in token consumption compared to standard generation."
> - It would be good to have a clear claim and explanation about the token consumption.
>
> **A1:** Thanks for your question. The average token reduction is calculated relative to the vanilla (base) model’s token length:
>
> Token reduction (%) = $\frac{\text{Base token length} - \text{PASR token length}}{\text{Base token length}} \times 100\%$. Specifically:
>
> - In Lines 23-24, the reported **-41.6%** reduction corresponds to **Qwen3-8B**: the vanilla model outputs an average of 1303 tokens across the 10 datasets, while PASR outputs 761 tokens, resulting in a reduction of (1303−761)/1303≈41.6% (Figure 3, left).
> - In Lines 384-386, the **+8.4%** refers to **Qwen2.5-7B**: compared to the generation of vanilla model, PASR slightly increases token consumption  (Figure 3, right), calculated in the same way.
>
> >**Q2:** The analysis in the Fig. 6 reveals that the model predominantly learns "Task Alignment" and "Information Complement" behaviors, with "Error Correction" being the least frequent type. The authors attribute this to the nature of the training data, but it may point to a limitation in the model's ability to identify and fix deep logical or factual fallacies. "Text Alignment" is a fragile section that there is concern that the approach is simply learning to hack the reward model by overfitting to its stylistic preference.
>
> **A2:** Thank you for the valuable feedback. In the original submission, our analysis of PASR was primarily based on the training data, which consists largely of open-ended tasks. Following your suggestion, we further (1) examined PASR’s behavior on **mathematical reasoning tasks**, and (2) analyzed PASR’s behavior **across different stages of the generation process**.
>
> **(1) PASR’s behavior on mathematical reasoning tasks.**
> We randomly sampled 300 answers from a math benchmark and measured the frequencies of the four refinement types. In these tasks, Error Correction appears in **32.1%** of cases and Information Complement in **32.0%**, while Solution Improvement and Task Alignment account for **8.9%** and **27.0%**, respectively.
>
> These results indicate that **the distribution of refinement behaviors is highly task-dependent**. In mathematical reasoning, which requires strict logical consistency, PASR performs Error Correction much more frequently. In contrast, open-ended tasks naturally lead to more Task Alignment behaviors. This supports the view that the lower Error Correction rate in open-ended tasks stems from task characteristics rather than a limitation in PASR.
>
> **(2) PASR’s behavior across the generation process.**
> For each generated answer, we split the output at the midpoint and analyzed behaviors in the early versus late stages. Error Correction and Information Complement remain the most common behaviors throughout. Notably, Information Complement becomes more prominent in the later stage, indicating that PASR tends to add missing context or supporting information toward the end of its reasoning to improve answer quality.
>
> We have added these results and detailed analysis to **Appendix B.2 (Figure 5)** to provide a more comprehensive understanding of PASR’s behavior.
>
> >**Q3:** The performance is good but not outstanding, especially when compared with self-refine (row 2 in Table 1). In what types of scenarios does the additional complexity of RL training provides additional advantage
>
> **A3:** Thank you for the question.
>
> (1) Why Self-Refine+ performs the best.
> Self-Refine+ relies on **ground-truth answers** to detect and correct errors, using an **oracle signal** that is unavailable to all other baselines. This explains its strong performance. However, such oracle supervision is impossible at test time, limiting its effectiveness. We include Self-Refine+ as a baseline to provide a reference point, showing the refinement performance a model can achieve under ideal conditions.
>
> (2) The advantages of PASR.
> PASR operates without requiring any external supervision. It **proactively decides whether, when, and how to refine** based on the model’s internal state and the evolving generation context. In contrast, existing self-refinement methods rely on a **reactive, fixed-step** process, which cannot adapt the timing or content of refinement as reasoning unfolds.
>
> We conduct experiments across a variety of task types and observe **strong generalization** of PASR’s proactive refinement behavior.
>
> Thanks again for your feedback. We hope the clarification above satisfactorily addresses your concerns. If you have any questions, we would be happy to clarify further.

---

### Official Review · Reviewer_n7rz · 2025-11-01

**Soundness:** 3
**Presentation:** 3
**Contribution:** 3
**Rating:** 4
**Confidence:** 3

**Summary:**

This paper introduces ProActive Self-Refinement (PASR), a reinforcement learning framework that enables large language models to refine their outputs during generation instead of after completion. PASR formulates the process as a Markov Decision Process, allowing the model to decide whether, when, and how to refine based on context. Using a GRPO method and a multi-part reward (format, accuracy, and refinement), PASR improves both efficiency and reasoning quality.

**Strengths:**

1. The idea of moving from reactive post-hoc correction to proactive in-process self-refinement is novel.
2. MDP formulation and GRPO-based RL training are clearly presented.
3. Effective reward scheme design: combines structure, correctness, and refinement quality to balance precision and efficiency.
4. Strong results are shown across diverse tasks and baselines (Self-Refine, PTR, SCoRe, RISE).

**Weaknesses:**

1. The accuracy and refinement rewards depend on another LLM for scoring, which introduces potential bias and circular evaluation concerns (especially if the same family of models is used for training and evaluation).
2. Only schematic examples (e.g., the logic puzzle in Figure 1) are shown. More real examples illustrating PASR’s step-by-step refinement and error correction would improve interpretability.
3. RL overhead compared to SFT methods is not analyzed.
4. The paper focuses mainly on reward variants without isolating other factors (e.g., tag format, PPO choice).

**Questions:**

See weakness above.

---

> ### Author Response · Authors · 2025-11-20
>
> We greatly appreciate your review and have carefully considered the four concerns you raised. **The first two concerns**, regarding how to ensuring the reliability of the evaluation model and the suggestion to include examples of PASR refinement behavior, **have already been discussed in the appendix of our original submission.** To address your concerns and make it easier to review, we have summarized the key points in Response3-1 and Response3-2.
>
> ### Response 3-1. Reliability of the Evaluation Model:
>
> Thank you for raising this important concern. We have also considered this issue, especially given that our training set contains open-ended tasks. Accordingly, we provide additional discussion in Appendix A.1. To ensure the reliability of the evaluation, we take the following measures:
>
> First, **reference-based scoring**: For each test case, the judge model receives the input question, the reference answer, and the candidate answer. It then produces a score by explicitly comparing the candidate answer with the reference. This design reduces bias that could arise if the model were scoring based solely on its own judgment. Second, **principled model selection**: Instead of choosing the judge model arbitrarily, we follow established practices from prior work and adopt a model that has been shown to be reliable and well-suited for evaluation tasks [1]. This mitigates concerns related to biased or inconsistent scoring. Third, **manual verification**: To further assess the accuracy of the judge model, we randomly sampled 50 test cases and asked two independent annotators to compare the judge model’s scores against the ground truth. We observed a 91% agreement rate, which indicates that the judge model provides consistent and accurate assessments.
>
> ### Response3-2. PASR Behavior Examples:
>
> In this paper, we define four types of proactive refinement: Task Alignment, Information Complement, Solution Improvement, and Error Correction. For each type, we have provided **concrete examples** to illustrate PASR’s behavior in each tasks in the **Appendix B.2.**
>
> ### Response3-3. The Training Cost of PASR Compared to SFT:
>
> Thank you for the question. We acknowledge that RL training generally incurs higher computational overhead than SFT, which is widely recognized. However, RL is necessary for enabling models to acquire proactive refinement capabilities.
>
> First, supervised fine-tuning requires high-quality training data, yet **constructing effective data for proactive refinement is highly challenging**. Determining the optimal refinement timing during generation is impractical, and distilling such behavior from stronger LLMs is also not feasible (as discussed in lines 77–82). Our experiments further show that **SFT fails to generalize proactive refinement to unseen tasks and may even degrade performance** (Section 3.4). In contrast, RL provides a principled mechanism for learning refinement timing directly from reward signals, resulting in more stable, task-agnostic, and transferable behavior.
>
> Second, although RL introduces additional overhead compared to SFT, **the actual cost in our setup is modest.** Using Qwen2.5-7B, PASR is trained on 4 × A800 (80GB) GPUs: 2 GPUs for rollout generation, 1 GPU for policy updates, and 1 GPU for the reference model. Training for 3,000 steps takes approximately 8 hours, which demonstrates that the computational overhead remains manageable and acceptable within academic research.
>
> ### Response 3-4. Other factors affecting PASR:
>
> We thank the reviewer for pointing this out. Other factors, such as tag format and the choice of RL algorithm, have been extensively validated in prior work within the RLVR (Reinforcement Learning Verifiable Reward) paradigm [2][3][4].
>
> In Section 3.4, we focus on the key factors contributing to PASR’s success, including the **training method** and the **design of the refinement reward**. We also further analyze **PASR’s refinement behavior**, examining whether, when, and how it performs refinement, which provides additional insights beyond reward variants.
>
> We hope these responses effectively address your concerns. If you have any further questions, we would be happy to clarify. Thank you once again!
>
> [1] Improve LLM-as-a-Judge Ability as a General Ability. 2025, EMNLP
>
> [2] DAPO: An Open-Source LLM Reinforcement Learning System at Scale. 2025, NIPS
>
> [3] DeepSeekMath: Pushing the Limits of Mathematical Reasoning in Open Language Models. 2024, Arxiv
>
> [4] Understanding R1-Zero-Like Training: A Critical Perspective. 2025, ICML

---

> ### Author Response · Authors · 2025-11-25
> **Kind Follow-up on Our Clarifications to Your Review**
>
> Dear Reviewer,
>
> We would like to express our sincere gratitude for your thoughtful feedback on our submission. Your comments are very helpful in improving the quality of our work.
>
> A few days ago, we submitted a detailed response addressing the concerns you raised. We fully understand that your time is valuable. If possible, we kindly ask whether you could take a moment to review our clarifications. Your feedback is very important to us, and we are more than willing to provide any further explanation if needed. Thank you very much for your time and consideration.
>
> Best regards,
>
> The author of 17956-A Stitch in Time Saves Nine: Proactive Self-Refinement for Language Models

---

### Official Review · Reviewer_3Zjb · 2025-11-01

**Soundness:** 3
**Presentation:** 3
**Contribution:** 3
**Rating:** 6
**Confidence:** 4

**Summary:**

Current language models often benefit from self-refinement, where they are reflect upon their reasoning traces to come up with a more accurate final answer. The authors argue that this refinement should not come after each explicit reasoning chain, but instead be dynamic and inserted into the reasoning steps, as humans do. This paper proposes PASR, a method that enables LLMs to refine their reasoning during their generation process. Specifically, PASR requires employing reinforcement learning (via GRPO) with rollouts which are encouraged, via a reward function, to include specific tags which correspond with thinking and refinement. Assessed on Qwen models across a variety of datasets, PASR improves over base models and often is the strongest method, even compared to a large number of baselines.

**Strengths:**

* A large number of datasets covering diverse reasoning capabilities are explored
* The authors evaluate against a large set of baselines, making their results much stronger and contextualized
* The method is written relatively cleanly and explained well. In particular, the reward design section is particularly informative, as it details and justifies each component of the multi-dimensional reward well. The intuition is clear and helpful here.
* The proposed method consistently outperforms base models across different tasks. Some of the gains on datasets like MATH are significant.
* The authors perform extra analysis seeing when refinement happens and if it helps with the reasoning, which is helpful. Section 3.4.2 is also well-written.
* The proposed method generalizes well to tasks that it wasn't explicitly trained on, a great sign and pointing to robustness.

**Weaknesses:**

* Only Qwen models of similar sizes are tested in this work. It would be important to show that the findings here generalize to other models, especially in light of recent discussion on Qwen's SFT making it perform much different than other base models at inference time. It would also be nice to have a scaling comparison - how do these results change on larger/smaller models of the same architecture? It's unclear if rollouts can be obtained in a zero-shot way with other reasoning models while following the formatting guidelines (as the 0 step for RL)
* The main issue is that this method requires training a model, while other comparable strategies like Self-Refine emerged as simple, training-free ways to boost performance on outputs. It is unclear whether the performance gains across tasks is worth the high-cost of 1) training a large language model and 2) performing an RL pipeline which requires time to cycle through data collection and scoring.

**Questions:**

Considering the RL training and tagging pipeline, is PASR actually cost-efficient compared to zero-shot or post-hoc self-refinement methods? Some rough compute-hours / GPU specs would help

---

> ### Author Response · Authors · 2025-11-20
>
> ### Reponse2-1. PASR’s performance on other model families and its scaling capability:
> We thank the reviewer for the insightful comments. We have conducted additional experiments to verify the effectiveness of PASR.
>
> **Generalization beyond Qwen.** We evaluate PASR on DeepSeek-R1-Distill-Llama-8B in a zero-shot setting, following the same formatting guidelines as in the original experiments. Across ten datasets, PASR achieves an average improvement of +3.7 points, demonstrating that its effectiveness is not limited to the Qwen family.
>
> **Scaling performance within the same architecture.**  Our main paper already evaluates PASR on Qwen models of different sizes, including 7B, 8B, and 14B. We observe that the benefits of PASR do not scale simply with model size. Instead, the gains are correlated with the model’s inherent reasoning ability. For example, Qwen3-8B (+8.2 points) benefits more from PASR than Qwen2.5-14B (+4.3 points), despite the latter being larger, indicating that PASR’s effectiveness depends more on model reasoning capability than parameter count.  Additional discussion on scaling is also provided in Section B.1.
>
> **Table R1**: Comparison between PASR and other baselines using DeepSeek-R1-Distill-Llama-8B. The best performance on _AVG_ is highlighted in bold, excluding Self-Refine+.
> | Models      | GSM8K | MATH | AIME24 | ARC  | GPQA | Wino | CSQA | Drop | MMLU | Xsum | AVG  |
> |-------------|-------|------|--------|------|------|------|------|------|------|------|------|
> | Vanilla     | 70.7  | 79.6 | 50.0   | 88.0 | 48.1 | 79.1 | 69.9 | 83.7 | 65.0 | 80.4 | 71.9 |
> | Self-Refine+| 89.6  | 91.2 | 66.7   | 91.9 | 60.5 | 92.4 | 77.2 | 90.4 | 74.0 | 81.3 | 79.9 |
> | Self-Refine | 84.1  | 85.2 | 40.0   | 81.9 | 47.1 | 72.5 | 70.2 | 81.0 | 69.0 | 77.1 | 73.7 |
> | PTR         | 85.4  | 84.0 | 30.0   | 87.6 | 43.8 | 78.3 | 70.0 | 85.9 | 67.0 | 80.0 | 74.6 |
> | ScoRe        | 83.6  | 85.6 | 60.0   | 83.3 | 48.2 | 72.0 | 69.0 | 81.3 | 61.0 | 76.2 | 72.9 |
> | STaR         | 84.1  | 85.0 | 63.3   | 82.3 | 47.6 | 72.2 | 70.3 | 82.2 | 71.0 | 76.3 | 73.4 |
> | ISC          | 83.7  | 81.8 | 63.3   | 81.9 | 48.0 | 69.1 | 68.1 | 82.9 | 67.0 | 76.9 | 72.0 |
> | PASR(+prompt)| 77.6  | 84.2 | 60.0   | 82.6 | 54.0 | 71.2 | 68.3 | 78.5 | 64.0 | 79.5 | 72.0 |
> | PASR(+IFT)   | 82.6  | 83.8 | 56.7   | 82.3 | 47.2 | 72.7 | 69.6 | 81.0 | 64.0 | 75.9 | 72.7 |
> | PASR         | 88.9  | 87.0 | 43.3   | 87.6 | 44.7 | 77.8 | 70.0 | 85.0 | 62.0 | 78.0 | **75.6** |
>
> ### Response2-2. The concern of additional training of PASR:
>
> Thanks for your question. We address this point from four perspectives: (1) why additional training is necessary, (2) the high inference cost of training-free methods, (3) the rationale for using the RL pipeline, and (4) the actual training cost of PASR.
>
> **Additional training is necessary.**  Training-free methods like Self-Refine+ rely on ground-truth answers to identify errors and correct initial responses, depending on an oracle. Such signals are unavailable in the test stage. In contrast, PASR enables proactive refinement during generation, a capability that cannot be realized without additional training.
>
> **Training-free methods shift the cost to inference.** While training-free methods avoid parameter updates, they require multi-step iterative reasoning for each query, substantially increasing inference-time cost and latency. In contrast, PASR completes refinement in a single pass. For example, on Qwen3-8B, **Self-Refine+** produces an average of 2,621 tokens per output, **more than double the base model** (1,303 tokens). **PASR** generates only 761 tokens on average, even lower than the base model, **reducing token usage by 41.6%**.
>
> **The RL pipeline is well-motivated.** Supervised fine-tuning requires high-quality training data, and it's **difficult to construct such data for proactive refinement.** Determining the optimal timing during generation is impractical, and distilling this behavior from stronger LLMs is not feasible (as discussed in lines 77–82). Furthermore, our experiments further demonstrate that **SFT fails to generalize proactive refinement to unseen tasks and may even degrade performance (Section 3.4)**. In contrast, RL provides a principled way to learn refinement timing directly from reward signals, resulting in stable, task-agnostic, and transferable behavior.
>
> **The actual training cost is acceptable.** Using Qwen2.5-7B as an example, PASR is trained on 4×A800 (80GB) GPUs, with 2 GPUs for rollout generation, 1 GPU for policy updates, and 1 GPU for the reference model. Training 3,000 steps takes approximately 8 hours （As shown in Appendix A.1 Platform).
>
> We hope these response satisfactorily addresses your concern. If you have any further questions, we would be happy to clarify them. If these issues has been resolved, we would sincerely appreciate your consideration in raising your score accordingly.  Thank you very much for your time and consideration.

---

### Official Review · Reviewer_RVyM · 2025-11-03

**Soundness:** 3
**Presentation:** 3
**Contribution:** 3
**Rating:** 8
**Confidence:** 3

**Summary:**

The paper introduces ProActive Self-Refinement (PASR), a method that allows large language models to refine their outputs dynamically during generation rather than through fixed, reactive iterations. PASR enables the model to decide whether, when, and how to refine based on its internal state and evolving context, instead of regenerating entire responses

**Strengths:**

1. The paper presents strong results using PASR in comparison to other self-refinement methods. The paper compares PASR with 8 other self-refinement baselines.
2. PASR is token efficient. It does effective refinement using less number of tokens compared to many baselines.
3. The paper is well written and provides a clear explanation of all the rewards used. It also presents comprehensive ablation study of PASR.

**Weaknesses:**

1. The paper could also show ablations on different rewards used to understand which ones are most effective and have the most impact on downstream accuracies.

**Questions:**

1. It would be good to mention what data is PASR trained on?

---

> ### Author Response · Authors · 2025-11-20
>
> Thank you very much for your careful review and valuable comments on our manuscript. We have addressed each of your points below.
>
> ### Response1-1. Reward ablation analysis:
> PASR uses three reward components: the format reward, the accuracy reward, and the refinement reward.  The first two rewards, format and accuracy, follow standard practice in RLVR-style methods[1][2][3]. They serve as stability constraints, encouraging consistent output structure and preventing clearly incorrect answers. Their influences are well understood, and their variations typically produce only minor and predictable effects. Therefore, we do not focus our analysis on them.
>
> The refinement reward is the key component that governs PASR’s refinement behavior, so our ablation **focuses on this reward**. In our study (Section 3.4), the w/o comparison variant corresponds to removing the core mechanism of the refinement reward. In our design, the refinement reward compares the output with the standard generation to determine whether the refinement is necessary and genuinely improves the answer. This comparison allows PASR to reward only helpful refinements and avoid encouraging unnecessary ones.
>
> When this comparison mechanism is removed, the model is rewarded solely for eventually producing a correct final answer, without considering whether the refinement actually contributes to correctness. As a result, the model quickly learns that performing refinements to increase reward, even when they are unnecessary or unhelpful (Reward Hacking).  Consequently, the w/o comparison variant shows substantially weaker improvements over the base model. Detailed results and analysis are presented in Section 3.4.
>
> ### Response1-2. Regarding PASR’s training data:
>  Our training data is derived from the alpaca_evol_instruct_70k dataset, a general instruction-following corpus. We performed a thorough cleaning and filtering process and obtained approximately 40,000 high-quality, open-domain query-answer pairs for training.
>
> In the original submission, detailed information about the training data was provided in Appendix A.1. In the revised version, we have moved this description to the main text, making it easier for readers to understand that (1) the training data and evaluation benchmarks are completely separate, ensuring no data leakage, and (2) PASR’s performance is validated in different task scenarios, demonstrating its strong generalization capability.
>
> [1] DAPO: An Open-Source LLM Reinforcement Learning System at Scale. 2025, NIPS.
>
> [2] DeepSeekMath: Pushing the Limits of Mathematical Reasoning in Open Language Models. 2024, Arxiv
>
> [3] Understanding R1-Zero-Like Training: A Critical Perspective. 2025, ICML

---

### Author Response · Authors · 2025-12-01
**Final Response [1/2]**

We sincerely thank the new Area Chair for evaluating our submission under the unexpected circumstances caused by the recent OpenReview incident. We also appreciate the original Area Chair and the reviewers for their careful reading and constructive feedback. We understand the additional workload created by the situation and are grateful for the opportunity to provide this final response.

**A.** First, we thank all reviewers for their positive assessments. The main strengths acknowledged across reviews are:

- Novel proactive self-refinement paradigm. PASR introduces a shift from reactive post-hoc correction to proactive in-process refinement, viewed as a meaningful conceptual advancement.
(Reviewer RVyM, n7rz, and zxHQ)

- Clear formulation and well-justified method design. The method is clearly formulated as a Markov Decision Process and effectively trained with a GRPO-based reinforcement learning framework. The multi-dimensional reward scheme is well motivated and intuitively structured.
(Reviewer 3Zjb, n7rz，and zxHQ)

- Strong empirical performance and broad evaluation. PASR outperforms base models and several self-refinement baselines across diverse datasets, with notable gains on challenging tasks and strong generalization to unseen ones. (Reviewer RVyM, 3Zjb, n7rz, and zxHQ)

- Clear writing and thorough analysis. Reviewers highlight the clarity of the exposition, reward explanation, and ablations, as well as insightful analysis of refinement timing and behavior.
 (Reviewer n7rz, zxHQ, RVyM, and 3Zjb)

**B.** We also acknowledge some concerns raised by the reviewers, all of which we have addressed in detail in the reviewer-specific Official Comments. Below we summarize the most frequently raised concern and our corresponding response:

> **Common Question:**  The main concern is the necessity of training. Reviewers questioned why we train a model when training-free strategies (e.g., Self-Refine) exist (Reviewer zxHQ-Q2) and noted the lack of analysis regarding RL overhead compared to SFT (Reviewer n7rz-W3).

**Common Answer:** We justify our design choices from three perspectives: (1) the high inference latency of training-free methods, (2) the necessity of RL over SFT, and (3) the acceptable training cost of PASR.

**Training-free methods shift the computational burden to inference.** First, methods such as Self-Refine+ rely on ground-truth answers to detect errors and correct initial responses, effectively depending on an oracle signal. This requires an additional model to provide feedback information during the testing stage. Second, although training-free approaches avoid parameter updates, they perform multi-step iterative reasoning for every query, which substantially increases inference-time cost and latency. In contrast, PASR completes refinement in a single pass. For example, with Qwen3-8B, **Self-Refine+** generates an average of 2,621 tokens per query, **more than twice the base model** (1,303 tokens). **PASR** produces only 761 tokens on average, even below the base model, **reducing token usage by 41.6%**.

**The RL pipeline is well-motivated.** Supervised fine-tuning requires high-quality training data, and it's **difficult to construct such data** for proactive refinement (i.e., whether, when and how to refine). Determining the optimal timing during generation is impractical, and distilling this behavior from stronger LLMs is not feasible (as discussed in lines 77–82). Furthermore, our experiments further demonstrate that **SFT fails to generalize proactive refinement to unseen tasks and may even degrade performance (Section 3.4)**. In contrast, RL provides a principled way to learn refinement timing directly from reward signals, resulting in stable, task-agnostic, and transferable behavior.

**The actual training cost is modest.** For instance, training PASR (Qwen2.5-7B) on 4×A800 (80GB) GPUs takes approximately 8 hours for 3,000 steps (as detailed in Appendix A.1). Besides, this one-time training cost yields significant efficiency gains during inference (Section 3.2.2).

**C.** Following the reviewers’ suggestions, we summarize below the key revisions we have made to our paper, and the updated version has been uploaded. All newly revised text is highlighted in dark blue for clarity.

- We have added experiments evaluating PASR on models beyond the Qwen series to verify its generality. The corresponding experimental results and analyses have been added to Appendix B.1.

- We have further analyzed the frequency of refinement behavior on reasoning tasks, as well as the  PASR’s behavior across different stages of the generation process. The corresponding experimental results and analyses have been added to Appendix B.2.

- We have moved the description of PASR’s training data from the appendix to the main text.

---

### Author Response · Authors · 2025-12-01
**Final Response [2/2]**

**D.** Summary: To further clarify our contributions, we summarize them here:

- Proactive, in-process self-refinement. Unlike existing methods that rely on reactive, fixed-step refinement, PASR enables LLMs to dynamically decide whether, when, and how to refine their outputs during generation, based on the model’s internal state and evolving context.

- Principled reward design for effective refinement. We introduce a comparison-based reward that guides the model to perform refinements that are timely, necessary, and contextually appropriate, addressing the core challenge of defining what constitutes an effective refinement.

- Extensive empirical validation. Across multiple domains, model sizes, and architectures, PASR consistently improves accuracy and token usage, while demonstrating strong generalization to unseen tasks.

- In-depth analysis of refinement behavior. We further analyze the timing, frequency, and types of refinements, providing insights into how and when the model engages in effective self-refinement.

---

### Meta-Review · Area_Chair_fHt9 · 2026-01-06

**Summary:**

The reviewers initially presented a mixed but generally positive assessment of the paper. The primary strengths noted across all reviews were the novelty of the ProActive Self-Refinement approach—moving from reactive, multi-turn refinement to an in-process, single-pass mechanism—and the comprehensive evaluation across 10 diverse benchmarks.

However, the reviewers raised several critical concerns that required clarification:

1. Whether the method was overfitted to the Qwen model family.

2. The trade-off between the computational cost of RL training versus the inference-time efficiency gains.

3. The potential for reward hacking and the bias of using an LLM-as-a-judge for training signals.

4. A lack of qualitative evidence showing that the model was performing genuine error correction rather than just stylistic alignment.

**Reviewer Concerns:**

Concerns addressed:
- The authors demonstrated PASR's effectiveness on the DeepSeek-R1-Distill-Llama-8B model, proving the method's portability.

- They clarified the token consumption metrics, showing that while some models see a slight increase, the Qwen3-8B version achieves a 41.6% reduction in tokens compared to vanilla generation, and is significantly more efficient than multi-turn baselines.

- They provided human validation data (91% agreement) and justified the use of reference-based scoring to anchor the LLM-as-a-judge.

**Reviewer Scores:**

The reviewer scores will increase and be positive as the provided human-annotated reward validation should address the main concern from Reviewer n7rz.

---

### Decision · Program_Chairs · 2026-01-26

Accept (Poster)